# Degradation of Tetracycline in Water by Fe-Modified Sterculia Foetida Biochar Activated Peroxodisulfate

Yuchen Zhang [1], Xigai Jia [1], Ziyang Kang [1], Xiaoxuan Kang [1], Ming Ge [1], Dongbin Zhang [2], Jilun Wei [2], Chongqing Wang [3,*] and Zhangxing He [1,4,*]

[1] College of Chemical Engineering, North China University of Science and Technology, Tangshan 063210, China
[2] Tangshan Sanyou Group Xingda Chemical Fiber Co., Ltd., Tangshan 063305, China
[3] School of Chemical Engineering, Zhengzhou University, Zhengzhou 450001, China
[4] Tangshan Sanyou Group Co., Ltd., Tangshan 063305, China
* Correspondence: cqwang1990@zzu.edu.cn (C.W.); zxhe@ncst.edu.cn (Z.H.)

**Abstract:** Tetracycline (TC) is a broad-spectrum antibiotic commonly, made use of in aquaculture and animal husbandry. After entering water bodies, it will represent a major threat to human health. In this study, sterculia foetida biochar (SFC) was readied by the combined hydrothermal pyrolysis (co-HTP) method with sterculia foetida as raw materials. Fen-SFC ($Fe_2$-SFC, $Fe_3$-SFC, and $Fe_4$-SFC) was obtained by doping SFH with different concentrations of $FeCl_3$. Finally, activation of peroxodisulfate (PDS) was achieved, using $Fe_3$-SFC to degrade TC. The degradation of TC obeyed pseudo-second-order kinetics, and the constant of the reaction rate was 0.491 L $mg^{-1}$ $min^{-1}$. Radical trapping experiments, EPR test and electrochemical tests evidenced that the high catalytic performance of the $Fe_3$-SFC/PDS system was ascribed to free radical pathway ($\bullet OH$ and $SO_4^{\bullet-}$) and non-radical pathway ($^1O_2$ and electron transfer), in which the latter plays a dominant role. This research not only demonstrates a new kind of biochar as an effective catalyst for PS activation, but also offers an avenue for the value-added reuse of sterculia foetida.

**Keywords:** peroxydisulfate activation; biochar; sterculia foetida; tetracycline wastewater; degradation

## 1. Introduction

Tetracycline (TC) as a broad-spectrum antibiotic has been diffusely made use of in clinical medicine and animal husbandry, and TC is usually discharged into the environment through medical wastewater and domestic wastewater [1–3]. Due to the poor biodegradability of TC, it may accumulate in the environment for a long time [4]. Many studies have indicated that after entering the aquatic environment [5], TC can cause the production of antibiotic resistance genes, boost bacterial resistance and cause ecotoxicity [6,7]. This will endanger human health and ecosystem stability [8]. As a result, an effective technique to degradation of TC in the water bodies is required [9,10].

Advanced oxidation processes (AOP) have shown great efficiency and almost non-selectivity in treating pollutants, when compared to traditional approaches [11,12]. A valid method for degrading organic contaminants in wastewater is the persulfate (PS)-based advanced oxidation process (AOP) [13,14]. Peroxymonosulfate (PMS) and peroxodisulfate (PDS) are two PS oxidants [15]. However, PS alone has a low degradation efficiency for antibiotics [16]. Various methods [17], such as photocatalytic [18], heating [19], UV light [20] andtransition metals [19], can generate reactive oxygen species (ROS) by activating PS, but these methods suffer from energy requirements, metal leaching, and other disadvantages. Research shows PS can be activated by carbon-based materials [21] such as carbon nanotubes (CNT) [22], activated carbon (AC) [23], biochar (BC) [24], and reduced graphene oxide (r-GO) [25], etc. Due to the advantages of abundant raw materials, low cost [26], rich mesoporous structure [27], and surface functional groups [28], BC produced from biochar waste has increasingly become a current research hotspot [23,29,30].

Modified BC doped with transition metals and heteroatoms is often utilized to prepare catalysts to improve the catalytic activity of BC [10,31,32]. Given Fe-based catalysts' high PS activation efficiency, combining biochar with iron-based materials to improve PS activation efficiency has received extensive attention [29]. As an activator, ferric chloride (FeCl3) [33] can enhance the accessible surface area of carbon materials while maintaining good electrical conductivity [34], promoting the adsorption of pollutants and electron transfer. Luo [29] et al. developed iron-rich magnetic BC (KMBC)-activated PS to efficiently degrade metronidazole, with a degradation rate of 98.4% after 120 min, and discovered that active ingredient content in magnetic BC [35], particularly the content of Fe, (II) was closely related to its activation properties [36,37].

BC was made in this work using sterculia foetida as a raw material [38,39]. The sterculia foetida was first subjected to hydrothermal treatment and then calcined to obtain SFC. Using FeCl3 as the precursor, iron-modified BC with different ratios (Fe2-SFC, Fe3-SFC, and Fe4-SFC) were further prepared. The degradation performance of the produced materials and SFC were compared when they were utilized to degrade TC. The catalysts' characteristics were examined by approaches for characterization [40] such as scanning electron microscopy (SEM), transmission electron microscopy (TEM), X-ray diffraction (XRD), Raman spectroscopy, BET surface area measurements, and X-ray photoelectron spectroscopy (XPS). The influences of certain factors (catalyst dosage, oxidizer dosage, solution temperatures, solution pH, concentration of initial, coexisting anions, etc.) on the removal of TC in Fe3-SFC/PDS system were probed. Finally, the mechanism of Fe3-SFC-activated PDS for TC degradation was brought forwards [41].

In this study, the Fe element is doped with sterculia foetida biochar to prepare iron-based biochar catalysts for the first time, which keeps carbon materials' high electrical conductivity and larger accessible surface area, and provides a more novel and efficient method for the value-added utilization of sterculia foetida. At the same time, the above materials are used as catalysts, and peroxydisulfate activation is used to achieve more efficient TC degradation under non-selective conditions and broaden the removal path of water pollutants.

## 2. Materials and Methods

### 2.1. Materials

Sterculia foetida was purchased from the supermarket. Absolute ethanol (purity > 99.7%), sodium hydroxide (NaOH, analytical reagent), sulphury acid ($H_2SO_4$, analytical reagent), sodium bicarbonate (NaHCO3, analytical reagent), and sodium phosphate (Na3PO4, analytical reagent) were obtained from Tianjin Yong Da Chemical Reagent Co., Ltd. Tetracycline (TC, analytical reagent) was purchased from Aladdin Industrial Corporation. Ethanol (EtOH, chromatographic grade) was acquired from China Shanghai Aladdin Biochemical Technology Co., Ltd. Anhydrous ferric chloride (FeCl3, analytical reagent) and Sodium persulfate (Na2S2O8, analytical reagent) were obtained from Shanghai Macklin Biochemical Co., Ltd. (Shanghai, China) Sodium nitrate Tianjin Bai Shi Chemical Industry Co., Ltd. (Tianjin, China) provided the analytical reagents Tertbutyl alcohol (TBA, $C_4H_{10}O$), sodium chloride (NaCl, analytical reagent), and sodium nitrate (NaNO3, analytical reagent). Alfa Esha (China) Chemical Co., Ltd. (Shanghai, China) supplied the furfuryl alcohol (FFA, $C_5H_6O_2$, purity > 98%).

### 2.2. Biochar-Based Catalyst Synthesis

Sterculia foetidas was soaked for 2 h in 1000 mL distilled water. The fruit cores were removed and filtered after the sterculia foetidas were soaked to expand, then diverted to five 100 mL autoclaves and heated for 18 h at 180 °C, and then cooled to room temperature. After washing and filtering the product with distilled water and hydrous ethanol, hydrothermal biochar (SFH) was obtained. It was placed inside a tubular reactor and calcined for 2 h at 800 °C in a $N_2$ environment to produce blank calcined biochar (SFC). After mixing and grinding the mass of SFH and FeCl3 in a certain ratio (1:2; 1:3 and 1:4), it was placed in

a tubular reactor to calcine the obtained material under the same conditions, and then soaked in 0.1 mM HCl solution for 12 h, according to the ratio of 0.5 g material to 200 mL hydrochloric acid solution. After filtration, the product was centrifuged and washed with distilled water to get Fe-doped FeCl₃-modified biochar (Fe₂-SFC; Fe₃-SFC and Fe₄-SFC). The material preparation method is shown in Scheme 1.

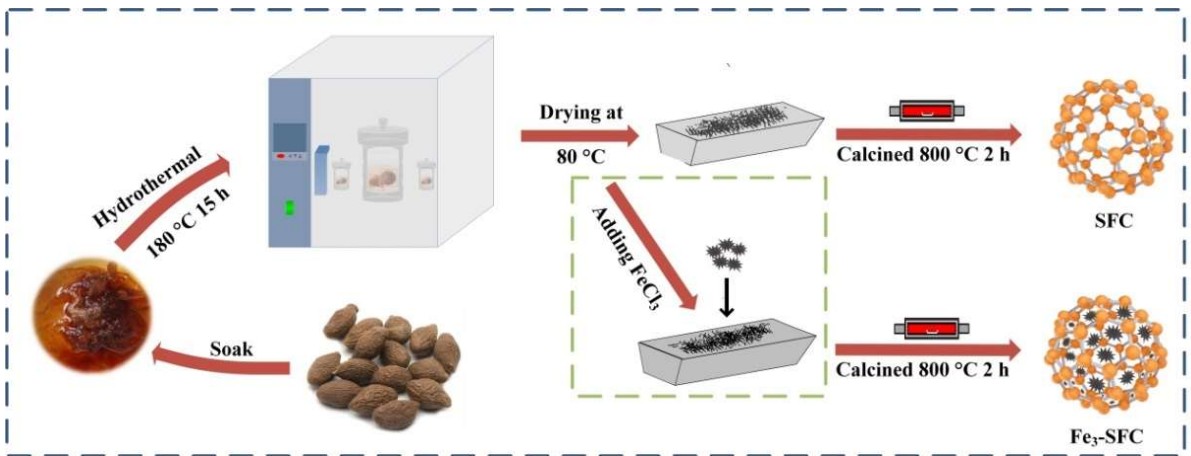

**Scheme 1.** Schematic representation of the preparation of SFC and Feₙ-SFC.

### 2.3. Characterization

Fourier transform infrared spectroscopy was used to identify the Fourier transform infrared (FT-IR, D-76275 Ettlingen Germany) of SFC and Fe₃-SFC as KBr (mass ratio of 1:400) pellets, before and after use. The D8-X-ray diffractometer (Bruker Company, Bremen, Germany) was used to find the obtained sample's XRD diffraction pattern. To examine the catalyst's morphology, a scanning electron microscope (SEM, JEOL, JSM-6390LV) was employed. The BET surface area, pore size, and pore volume of the catalyst were determined using a specific surface area pore size analyzer (Beijing Beishi Shi De Instrument Technology Co., Ltd., 3H2000PM1 Beijing China). X-ray photoelectron spectroscopy (Thermo KAlpha XPS America) was used to determine elemental valence of the catalyst. An electron spin resonance spectrometer (EPR) was used to detect the free radicals produced in the BC/PDS system.

### 2.4. Catalytic Degradation Experiment

In a typical method, 10 mg of biochar catalyst and 100 mL of 20 mg/L TC solution were added into a 250 mL beaker, and ultrasonic treatment was carried out for 1 min. It was quickly transferred to a water bath, stirred at 400 r/min, and 100 mg of PDS was added to the beaker solution to start the degradation reaction after 15 s of stirring. A certain amount of the sample solution (3 mL) was removed in a given time (0, 1, 2, 5, 10, 20, 40, 60 min), and filtered with 0.45 μm polyethersulfone membrane. The absorbance of the filtrate was immediately detected by UV-V spectrophotometer at λ = 359.5 nm.

Without special instructions, all experiments were performed under the conditions of temperature 25 °C, pH 6.93, and initial TC concentration of 20 mg/L, with no adjustment required. The effects of different kinds of biochar on TC degradation were studied by using the above method, and the first-order and second-order kinetics analyses were made. Different biochar concentrations (0.05 g/L, 0.10 g/L and 0.15 g/L), different PDS concentrations (0.5 g/L, 1.0 g/L and 1.5 g/L), different pH values (3.01, 6.93 and 11.03), different TC concentrations (10 mg/L, 20 mg/L and 30 mg/L), different temperatures (25 °C, 35 °C and 45 °C) and different anions (Cl⁻, NO₃⁻, HPO₄²⁻, HCO₃⁻) of the TC degradation were investigated. The free radical capture experiment and carbon cycle experiment in the process of TC degradation were tested.

### 2.5. Electrochemical Test

Preparation of the working electrode was as follows: 20 mg biochar was combined with 2 mL ethanol for 3 h of ultrasonic treatment, then dropped on the glassy carbon electrode, dried for 10 min at 60 °C, and then left to stand at 25 °C for 4 h. Platinum wire electrode was a counter-electrode, saturated calomel electrode was the reference electrode, with 50 mM $Na_2SO_4$ as the electrolyte.

Linear sweep voltammetry (LSV), cyclic voltammetry (CV) and electrochemical impedance spectroscopy (EIS) were implemented at the electrochemical workstation (Chi660e, Shanghai CH Instrument Technology Co., Ltd.). The LSV diagram of SFC under different conditions (separate SFC, SFC + PDS and SFC + PDS + TC) was measured at 0.05 V/s sweep speed and −1.0–1.5 V potential. At the same time, the LSV of $Fe_3$-SFC was measured at the same method and experimental conditions. At −1.5~1.5 V potential and 0.005 V/s scanning speed, the cyclic voltammetry (CV) curves of SFC and $Fe_3$-SFC were measured. EIS was measured with an amplitude of 0.005 V and a range of $10^{-1}$~$10^{5}$ Hz in frequency.

## 3. Results and Discussion

### 3.1. Characterization

The changes in surface functional groups before and after the reaction of SFC and $Fe_3$-SFC are shown in Figure 1a,b using FT-IR analysis. C-H (610 cm$^{-1}$) [42], C-OH/C-O-C (1100 cm$^{-1}$) [43], C-N (1380 cm$^{-1}$) [44] and -OH/-NH$_2$ (1570, 3420 cm$^{-1}$) [45] are the main tensile vibration functional groups of SFC, and these functional groups also react with $Fe_3$-SFC. The stretching vibration frequency of C-OH/C-O-C in $Fe_3$-SFC is higher than in SFC, whereas the stretching vibration of C-N, and -OH/-NH$_2$ is more noticeable in SFC. The bands in SFC are slightly different from $Fe_3$-SFC in the FT-IR spectra [46], and the functional groups on the surface of the two materials changed before and after the reaction, indicating that these surface functional groups make an important impact in catalytic degradation [47,48]. The functional groups on its surface will react with PDS, and its main reaction equation is [3]:

$$C\text{-}OH + S_2O_8^{2-} \rightarrow C\text{-}O\bullet + SO_4^{\bullet-} + HSO_4^{-} \tag{1}$$

$$C\text{-}O + S_2O_8^{2-} \rightarrow C\text{-}O\bullet + 2SO_4^{\bullet-} \tag{2}$$

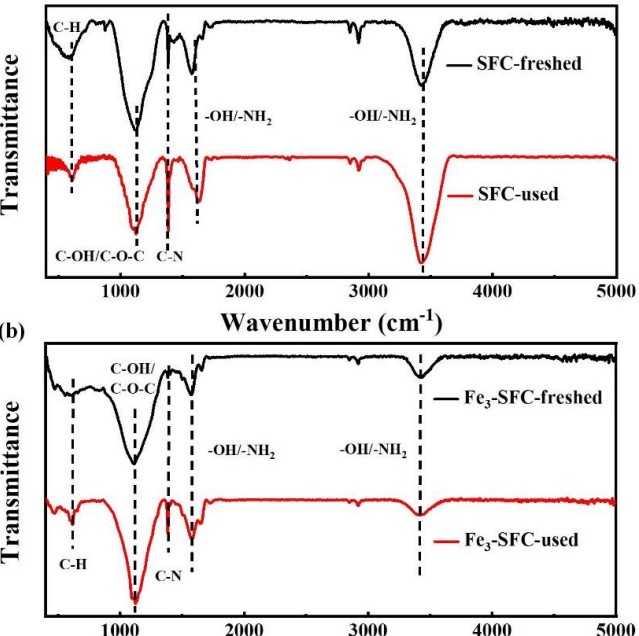

**Figure 1.** *Cont.*

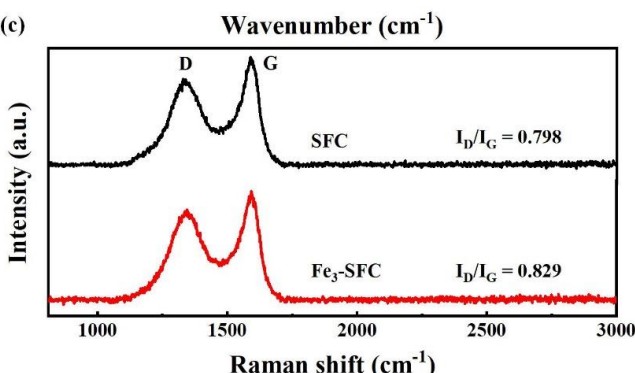

**Figure 1.** FT-IR spectra of the two materials before and after TC adsorption (**a**,**b**); Raman spectra of SFC and Fe$_3$−SFC (**c**).

Figure 1c shows the Raman spectra of SFC and Fe$_3$-SFC. The D and the G bands of the carbon material are represented by the peaks at 1350 and 1590 cm$^{-1}$ [49], respectively. SFC and Fe$_3$-SFC have I$_D$/I$_G$ values of 0.798 and 0.829, respectively, indicating that Fe$_3$-SFC has more defects than SFC, which can supply more defect sites for PDS to activate the catalytic performance of biochar [50,51].

SEM images of SFC and Fe$_3$-SFC are shown in Figure 2a–d. Although both SFC and Fe$_3$-SFC are composed of granular structures, there are significant differences in material morphology between the two catalysts [52]. Fe$_3$-SFC has larger particles and irregular surface roughness due to the incorporation of a small amount of Fe, while SFC exhibits relatively regular small particle shapes with smooth and more regular surfaces. As shown in Figure 2e–h, Fe$_3$-SFC contains C, N, O and Fe elements, the distribution is relatively uniform, and the content of Fe element is relatively low. Figure 2i,g again shows that both SFC and Fe$_3$-SFC contain the elements C, N and O, and that Fe$_3$-SFC also contains the Fe element.

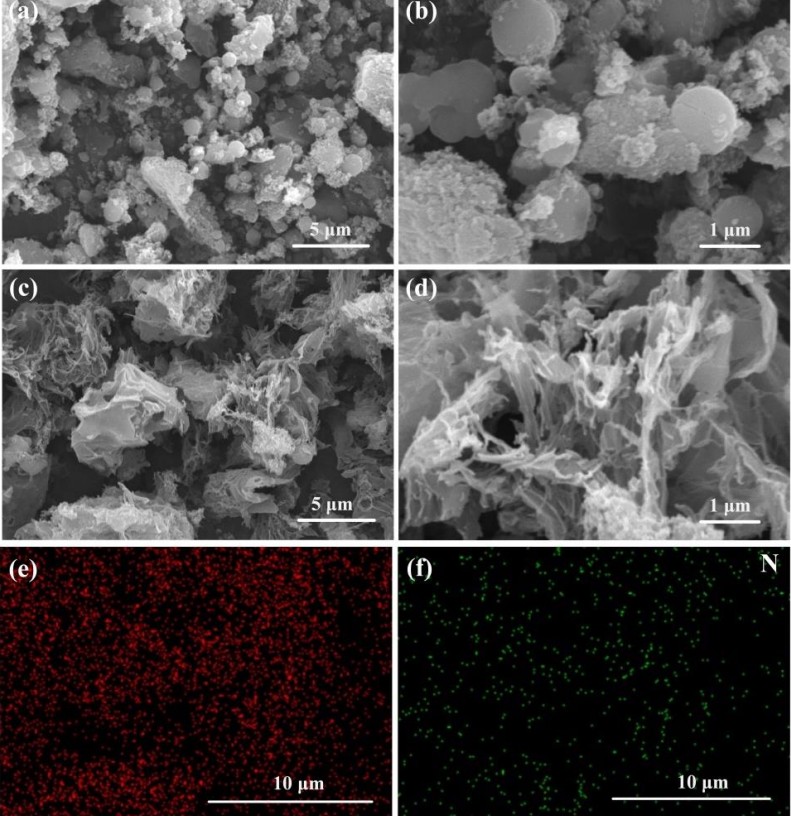

**Figure 2.** *Cont.*

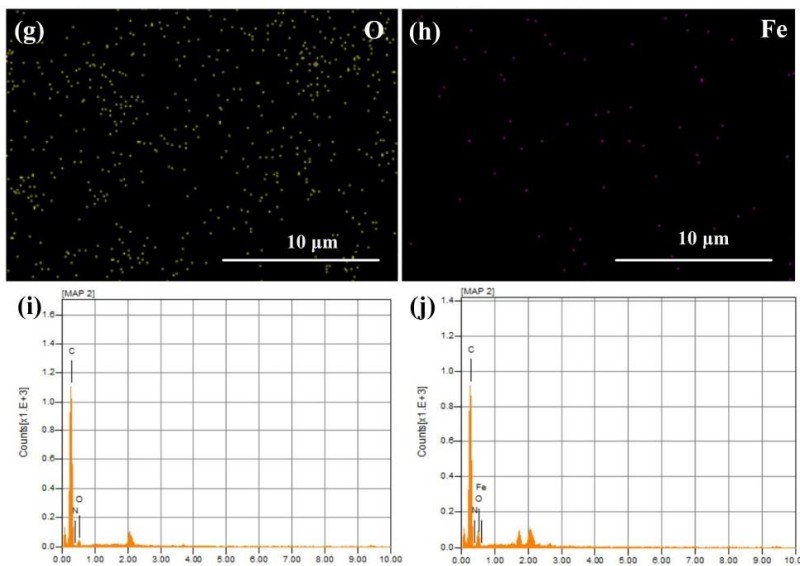

**Figure 2.** SEM images of SFC (**a**,**b**) and Fe$_3$-SFC (**c**,**d**); EDS mapping of Fe$_3$-SFC (**e**–**h**); EDS spectra of SFC (**i**); EDS spectra of Fe$_3$-SFC (**j**).

The crystal structure of the obtained carbon catalyst was investigated using XRD measurements. As shown in Figure 3, the diffraction peaks of SFC, Fe$_3$-SFC and Fe$_3$-SFC used at 2θ are at 20.8°, which corresponds to the (002) graphite structural plane [53]. At the same time, no other peaks were added after the modification of Fe, which indicated that most of Fe did not remain on the surface or inside of biochar, and Fe changed the structure of biochar during calcination. The peaks of Fe$_3$-SFC did not change significantly after use, which proves that Fe$_3$-SFC has good stability.

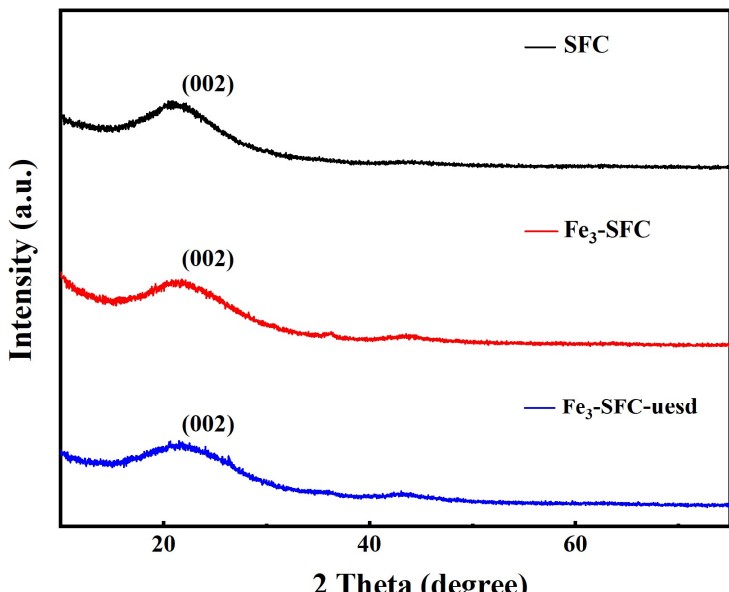

**Figure 3.** XRD patterns of SFC; Fe$_3$-SFC and Fe$_3$-SFC after use.

The N$_2$ adsorption and desorption isotherms of SFC and Fe$_3$-SFC are demonstrated in Figure 4a. SFC and Fe$_3$-SFC are shown as typical type II isotherms [54]. Fe$_3$-SFC can offer more active sites for the activation of PDS, since both its BET surface area and pore volume are greater than those of SFC (Table 1) at 959.2110 m$^2$/g and 1.0152 cm$^3$/g, respectively (Table 1) [52,55]. Figure 4b shows the pore size–distribution curves of SFC and Fe$_3$-SFC. Transition pores and micropores are found in both SFC and Fe$_3$-SFC [56].

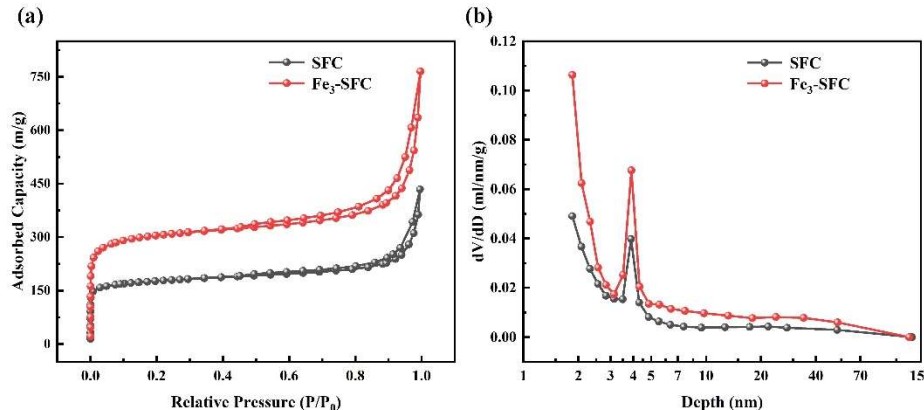

**Figure 4.** $N_2$ adsorption–desorption isotherms (**a**); and pore–size distribution curve (**b**).

**Table 1.** Specific surface area and pore volume of SFC and $Fe_3$-SFC.

| Sample | Surface Area ($m^2/g$) | Pore Volume ($cm^3/g$) |
|---|---|---|
| SFC | 558.5050 | 0.5649 |
| $Fe_3$-SFC | 959.2110 | 1.0152 |

The chemical composition and morphology of SFC and $Fe_3$-SFC were analyzed using XPS, shown in Figure 5. According to XPS results, both SFC and $Fe_3$-SFC contain C, N, and O elements, and $Fe_3$-SFC also contains Fe (Figure 5a). The atomic content ratios of C, N, O and Fe in SFC and $Fe_3$-SFC are shown in Figure 5b. It shows that the C content of $Fe_3$-SFC after Fe doping is relatively lower than that of SFC. Meanwhile, the content of N and O increases, which provides more functional groups, and the Fe element is also doped in it, but its content is less than 5%. The C atom content of SFC is higher than that of $Fe_3$-SFC, while N and O are lower than $Fe_3$-SFC. C1s can be deconvolved into distinct peaks, as seen in Figure 5c. Three distinctive peaks in SFC and $Fe_3$-SFC are attributed to C-C/C=C, C-H, and C-OH bonds, respectively, at energies of 284.8, 285.5, and 286.4 eV [53]. The N1s spectra of the two materials can be deconvoluted into three peaks (Figure 5d). These characteristic peaks at 401.09, 399.8, and 398.4 eV can be attributed to graphitic N, pyrrolic N, and pyridine N, respectively [57], and the major peaks for both are graphitic N. Figure 5e shows the XPS spectra of O1s for the two materials. The O1s XPS spectrum of SFC is divided into three peaks: C-OH (531.6 eV), O-H (533.7 eV), and C=O (532.7 eV). $Fe_3$-SFC has one more peak than SFC at 530.4 eV, which is Fe-O [58]. In addition, Figure 5f shows the Fe 2p spectrum, in which the two peaks at roughly 713.7 and 725.5 eV, correspond to $Fe^{3+}$ $2p_{3/2}$ and $Fe^{3+}$ $2p_{1/2}$ in $Fe_3$-SFC [55]. The peak at 711.6 eV belongs to the $2p_{1/2}$ peak of $Fe^{2+}$ [59]. The two peaks at 723.2 and 718.5 eV can be ascribed to $Fe^{2+}$ and Fe $2p_{3/2}$, respectively, indicating the presence of $Fe^0$ species on the $Fe_3$-SFC surface [58].

*3.2. TC Degradation*

The TC adsorption and catalytic properties of SFC and $Fe_n$-SFC were investigated. When comparing Figure 6a,b, it can be shown that after adding PDS, both SFC and $Fe_n$-SFC have a greater removal effect. In the instance of PDS, the TC degradation was investigated. After 60 min, the degradation effect of SFC on TC was very limited, at only 19.2%. However, in the $Fe_n$-SFC/PDS system, except $Fe_4$-SFC, the TC degradation rate of the other modified catalysts could reach more than 75.0% (Figure 6b). The findings demonstrate that adding $FeCl_3$ considerably improved the capacity to remove TC. In a certain range, the removal rate increased as the $FeCl_3$ level increased. When $FeCl_3$:SFC = 3:1, the highest degradation efficiency can reach 91.5%. When $FeCl_3$:SFC = 4:1, the degradation of catalytic performance is even less than that of blank biochar. This indicates that the amount of $FeCl_3$ has reached saturation, and when $FeCl_3$:SFC = 3:1, excessive $FeCl_3$ may take up reaction sites on the

biochar surface and reduce the degradation efficiency. It proved that $FeCl_3:SFC = 3:1$ is the best ratio to remove TC.

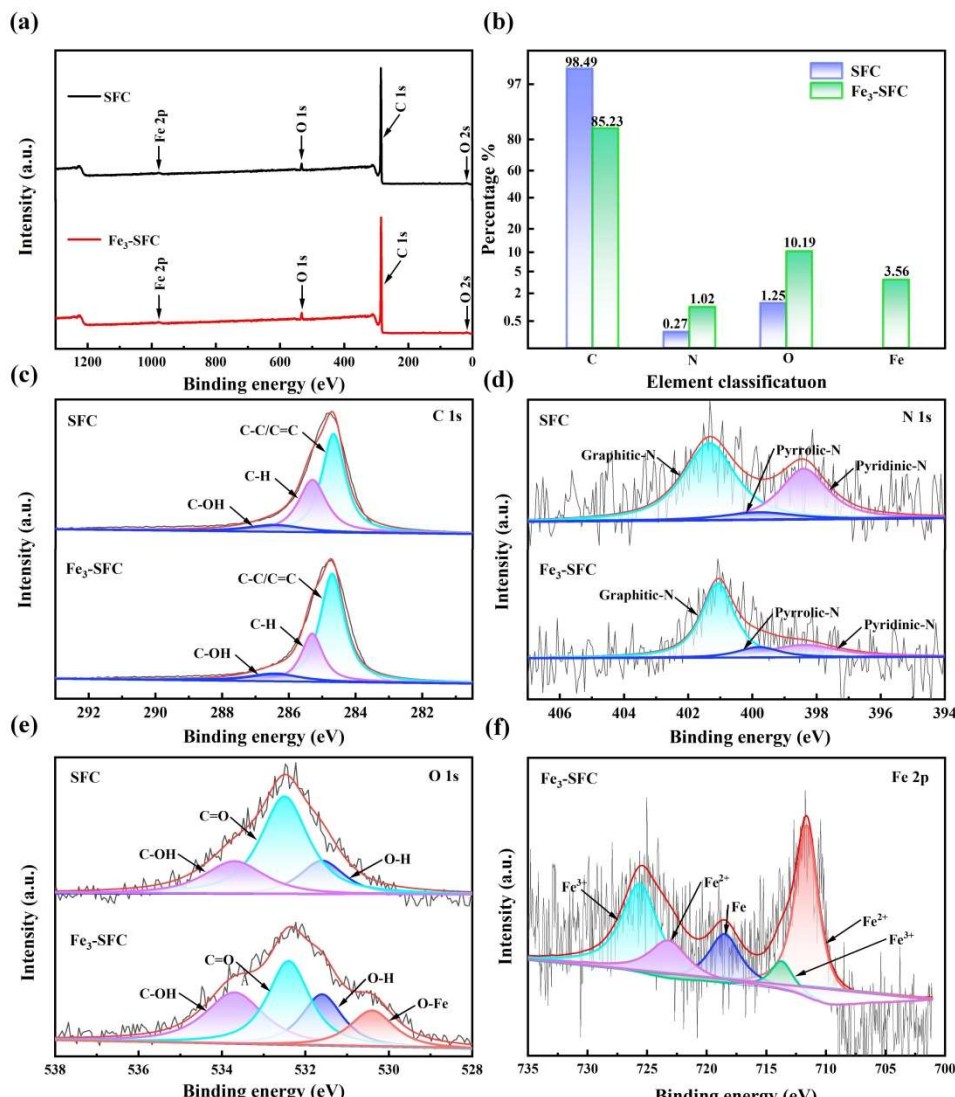

**Figure 5.** XPS survey scan of SFC and $Fe_3$-SFC (**a**); proportion of C, N, O, and Fe content (**b**); C1s peaks of SFC and $Fe_3$-SFC (**c**); N1s peaks of SFC and $Fe_3$-SFC (**d**); O1s peaks of SFC and $Fe_3$-SFC (**e**); Fe2p peaks of SFC and $Fe_3$-SFC (**f**).

Characterization of TC degradation in $Fe_3$-SFC/PDS systems using pseudo-first-order and pseudo-second-order kinetics (Equations (1) and (2)) [60,61].

$$\ln(C_0/C_t) = k_1 t \qquad (3)$$

$$1/C_t - 1/C_0 = k_2 t \qquad (4)$$

TC concentrations at time 0 and t are represented by $C_0$ and $C_t$. The first- and second-order rate constants are denoted by $k_1$ and $k_2$, respectively. Equations (1) and (2) were used to monitor the TC-degrading kinetic model in the $Fe_3$-SFC/PDS system (Figure 6c,d). The second-order model fitting's correlation coefficient ($R^2$) is 0.998, greater than the first-order model's (0.924) (Table 2). This outcome proves that the second-order reaction kinetics are better suited for TC degradation in the $Fe_3$-SFC/PDS system [61].

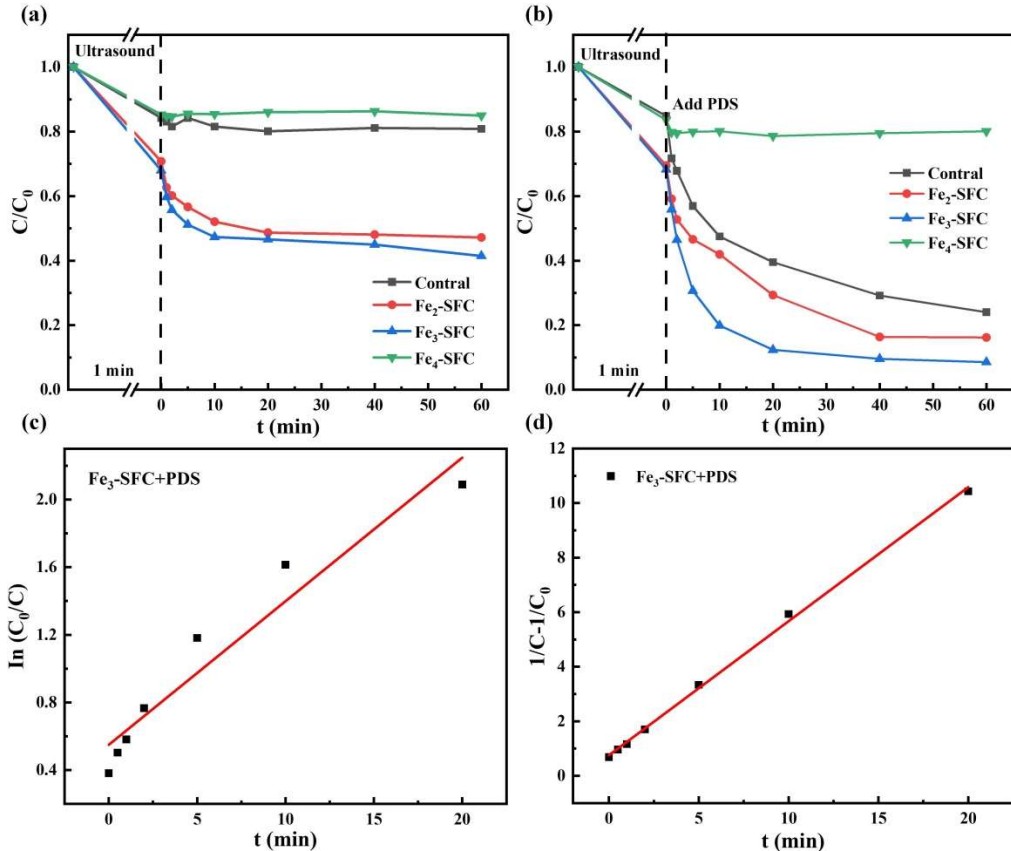

**Figure 6.** TC adsorption by the different biochar (**a**); TC degradation by the different biochar (**b**) the pseudo-first-order kinetic fitting curve (**c**); and the pseudo-second-order kinetics fitting curve (**d**).

**Table 2.** Fitting results regarding the degradation of TC in $Fe_3$-SFC/PDS system.

| Fitting Formula | Dynamical Equation | k | $R^2$ |
|---|---|---|---|
| Pseudo-first-order kinetic | $-\ln(C_0/C_t) = 0.085t + 0.549$ | 0.085 $min^{-1}$ | 0.924 |
| Pseudo-second-order kinetic | $1/C_t - 1/C_0 = 0.491t + 0.756$ | 0.491·L $mg^{-1}\cdot min^{-1}$ | 0.998 |

Figure 7a shows that $Fe_3$-SFC dosage had a favorable effect on TC degradation, as degradation rate of TC also increased from 81.9% to 92.3% as the amount of $Fe_3$-SFC was enhanced from 0.05 g/L to 0.15 g/L. This suggested that high doses of $Fe_3$-SFC could effectively improve the removal rate of TC, as more $Fe_3$-SFC could certify more active sites for activating PDS, thereby increasing the activation efficiency [62]. Figure 7b depicts the result of PDS concentration on degradation of TC. With an increase in PDS concentration, it was seen that the rate of TC degradation decreased, probably because excessive PDS would react with active oxidizing species to form some substances with low oxidation performance. But the amount of PDS had little impact on the catalytic degradation performance of the $Fe_3$-SFC/PDS system [63]. Figure 7c shows that, as the initial concentration of TC increases, its degradation rate decreased, which indicated that active sites on the $Fe_3$-SFC surface are more easily occupied by TC. Therefore, the contact between PDS and $Fe_3$-SFC is reduced [64].

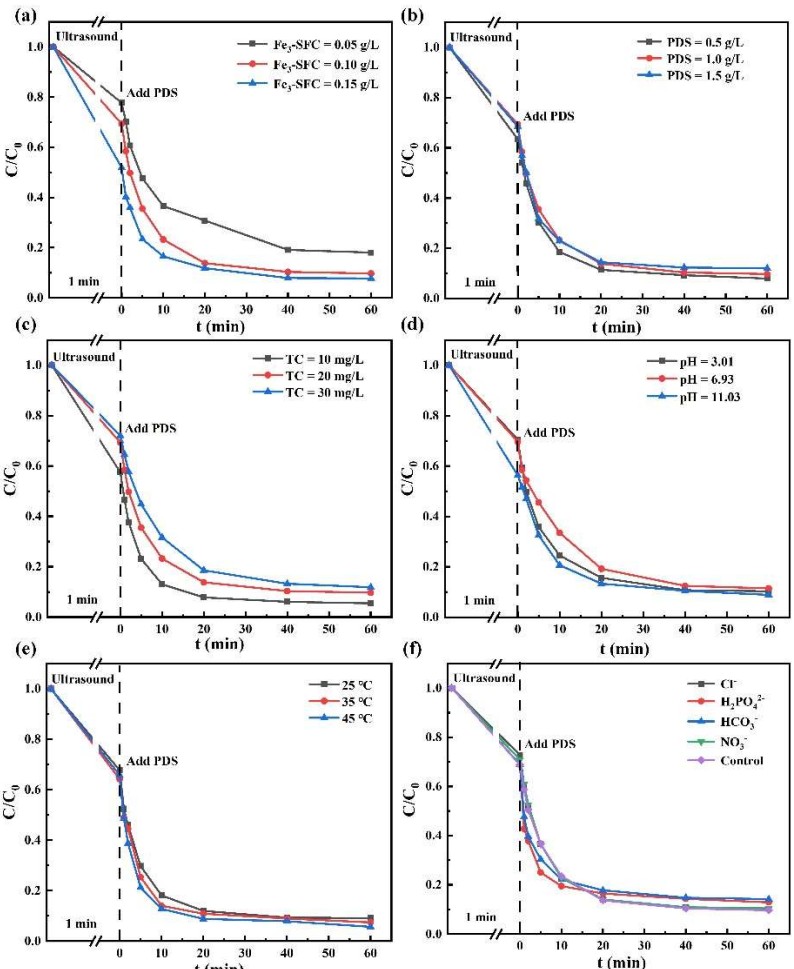

**Figure 7.** The influence of catalyst dosage (**a**); PDS dosage (**b**); TC concentrations (**c**); initial pH (**d**); temperature of reaction (**e**) and anions (**f**) on the degradation of TC in the $Fe_3-SFC/PDS$ system.

Within the $Fe_3$-SFC/PDS system, the effect of initial pH ranging from 3.01 to 11.03 on TC degradation is shown in Figure 7d. Over a pH range of 3.01 to 11.03, nearly overlapping TC degradation curves were obtained, indicating that $Fe_3$-SFC was catalytically stable over a wide working pH range [65,66]. When temperature increased from 25 °C to 35 °C, TC degradation rate increased slightly, but was almost unchanged (Figure 7e), which showed that higher temperature could enhance the catalytic performance of the $Fe_3$-SFC/PDS system, which was due to the higher temperature. High temperature could activate PDS to produce more active species, but the effect of this change was relatively weak [61]. Figure 7f shows the effects of $Cl^-$, $HPO_4^{2-}$, $HCO_3^-$, and $NO_3^-$ on the degradation efficiency of TC. The existence of four anions showed different degrees of inhibition on the degradation of TC. $Cl^-$ and $NO_3^-$ had only a minor inhibitory effect. $Cl^-$ can react with $SO_4^{\bullet-}$ to form $Cl^\bullet$ with relatively weak oxidizing power [67], and $NO_3^-$ possessed a quenching effect on $\bullet OH$ [61]. Relatively strong inhibition of TC degradation by $HPO_4^{2-}$ and $HCO_3^-$. $HCO_3^-$ had a powerful removal effect on $SO_4^{\bullet-}$ and $\bullet OH$, thus decreasing TC degradation efficiency [68]. $HPO_4^{2-}$ in-solution was also the scavenger of $\bullet OH$ and $SO_4^{\bullet-}$ [69]. In addition, $HPO_4^{2-}$ could form complexes with surface metal ions on $Fe_3$-SFC/PDS, and the active sites on the catalyst surface would be covered by complexes [70], thus hindering the TC degradation.

$$Cl^- + SO_4^{\bullet-} \rightarrow Cl^\bullet + SO_4^{2-} \tag{5}$$

$$NO_3^- + \bullet OH \rightarrow NO_3^\bullet + OH^- \tag{6}$$

$$HCO_3^- + SO_4^{\bullet-} \rightarrow HCO_3^\bullet + SO_4^{2-} \tag{7}$$

$$HCO_3^- + \bullet OH \rightarrow H_2O + CO_3^{\bullet -} \tag{8}$$

$$HPO_4^{2-} + SO_4^{\bullet -} \rightarrow HPO_4^{\bullet -} + SO_4^{2-} \tag{9}$$

$$HPO_4^{2-} + \bullet OH \rightarrow H_2O + HPO_4^{\bullet -} \tag{10}$$

The reusability of $Fe_3$-SFC was investigated (Figure 8). Three recovery experiments were performed on $Fe_3$-SFC. After each experiment, recovery, washing and drying of the spent N-BC catalyst were performed. It could be seen that degradation of TC in three runs were 90.5%, 83.9% and 51.4% after 60 min of reaction, respectively. With the increase of recycling times, TC degradation gradually decreases, which might be due to reduction of surface functional groups and coverage of active sites by intermediates, e.g., [66], which indicated that $Fe_3$-SFC had a certain lifetime before being consumed in the reaction. However, the degradation rate after three recycling cycles was 51.4%, indicating that $Fe_3$-SFC still had the ability to remove TC from the aquatic environment [44].

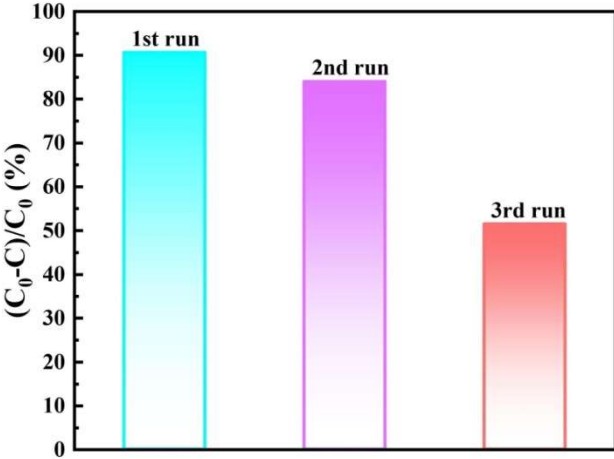

**Figure 8.** Reusability experiment of $Fe_3$-SFC.

Table 3 compares the degradation of pollutants by $Fe_3$-SFC and other biochar catalysts.

**Table 3.** Comparison of $Fe_3$-SFC and other materials in terms of pollutant degradation.

| Catalysts | Target Pollutants | Catalyst Dosage (G/L) | Pollutant Concentration (Mg/L) | Persulfatedosage (G/L) | Reaction Time | Degradation Conditions | D.E. (%) | Ref. |
|---|---|---|---|---|---|---|---|---|
| $Fe_3$-SFC | Tetracycline | 0.1 | 10 | 0.4 | 1 h | pH: 6.93 T: 25 °C | 91.50 | This work |
| KOH activation biochar | Tetracycline | 0.2 | 50 | 2 | 3 h | pH: 7 T: 25 °C | 82.36 | [66] |
| Goethite/ biochar | Tetracycline | 0.5 | 20 | 0.1 | 1 h | pH: 7 T: 25 °C | 66.71 | [64] |
| N-doped Enteromorpha prolifera-derived magnetic biochar | Tetracycline | 0.2 | 50 | 0.4 | 5 h | pH: 4 T: 25 °C | 87.2 | [71] |
| Passion fruit shell-derived biochar | Tetracycline | 0.4 | 20 | 0.1 | 2 h | pH: 7 T: 25 °C | 90.91 | [61] |
| Black fungus-derived N-doped biochar | Tetracycline | 0.3 | 20 | 0.4 | 1 h | pH: 6.9 T: 25 °C | 89.8 | [53] |
| Fe(II)-rich potassium-doped magnetic biochar | Metronidazole | 0.5 | 20 | 10 mM | 120 min | pH: 6.5 T: 25 °C | 98.4 | [72] |
| Fe, N co-doped biochar (Fe-N-BC) | Acid orange | 0.2 | 20 | 1 mM | 40 min | pH: 3 T: 25 °C | 100 | [57] |

**Table 3.** *Cont.*

| Catalysts | Target Pollutants | Catalyst Dosage (G/L) | Pollutant Concentration (Mg/L) | Persulfatedosage (G/L) | Reaction Time | Degradation Conditions | D.E. (%) | Ref. |
|---|---|---|---|---|---|---|---|---|
| Composite of iron sulfide and biochar (FeS@BC) | Tetracycline | 0.3 | 200 | 10 mM | 30 min | pH: 3.6 T: 25 °C | 87.4 | [73] |
| Magnetic rape straw biochar (MRSB) | Tetracycline hydrochloride | 1 | 20 | 8 mM | 120 min | pH: 5.68 T: 25 °C | 98.02 | [74] |
| Magnetic biochar was prepared from dewatered piggery sludge | Tetracycline | 0.75 | 6.7 | 20 mg/L | 120 min | pH: 7 T: 25 °C | 66.87 | [75] |
| Biochar supported nanosized iron (nFe(0)/BC) | Tetracycline | 0.4 | 100 | 1 mM | 240 min | pH: 5 T: 25 °C | 97.68 | [76] |

### 3.3. Reaction Mechanism

To better grasp degradation mechanism of TC and discern active substances in the system, quenching experiments were performed on the $Fe_3$-SFC/PDS system. Ethanol (EtOH) had been reported to be a typical •OH and $SO_4^{•-}$ trap, and *t*-butanol (TBA) was supposed to be able to capture •OH [77,78]. Singlet oxygen ($^1O_2$) could be largely trapped by furfuryl alcohol (FFA) [79]. Figure 9a shows that TC degradation efficiency was significantly reduced in the presence of FFA, whereas the degradation efficiency of TC decreased only slightly in the presence of TBA or EtOH. This result demonstrated that the $Fe_3$-SFC/PDS system produced a large amount of $^1O_2$ and a bit of •OH and $SO_4^{•-}$, indicating that main active species for degrading TC was $^1O_2$. In order to further confirm existence of $^1O_2$ in this system, it was measured using EPR technique. A strong signal of TEMP $^1O_2$ was heeded in the $Fe_3$-SFC/PDS-TEMP system, as shown in Figure 9b, indicating the presence of $^1O_2$ in the $Fe_3$-SFC/PDS-TEMP system [61].

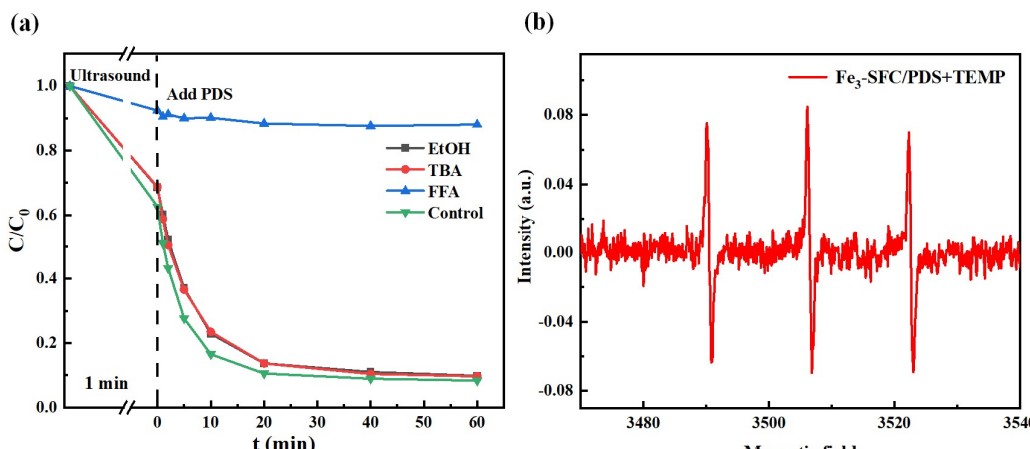

**Figure 9.** Effects of different quenchers on TC degradation in $Fe_3$−SFC/PDS system (**a**) and the EPR spectrum in $Fe_3$−SFC/PDS system with TEMP (**b**).

Figure 10 shows the LSV curves, CV curves, and EIS analysis of SFC and $Fe_3$-SFC under different conditions. The conduction of biochar could be clarified by electrochemical measurement [80]. Through the LSV diagrams of the two materials (Figure 10a,b), it could be found that the current increases significantly [81] with the increase of PDS solution. This indicates the interaction and electron rearrangement between PDS and biochar-based catalysts [82]. It is worth noting that injecting TC afterward caused another current increase, demonstrating that electron transfer [83] was faster in the PDS/SFC or $Fe_3$-SFC/TC ternary system. The birth material carbon was thought to act as a bridge for the formed current, promoting electron transfer from the TC molecule to the PDS [84]. When the currents of the

two systems were compared, it could be said that the current of Fe$_3$-SFC was significantly higher than that of SFC, indicating that Fe$_3$-SFC had a better electron transfer ability than SFC [85].

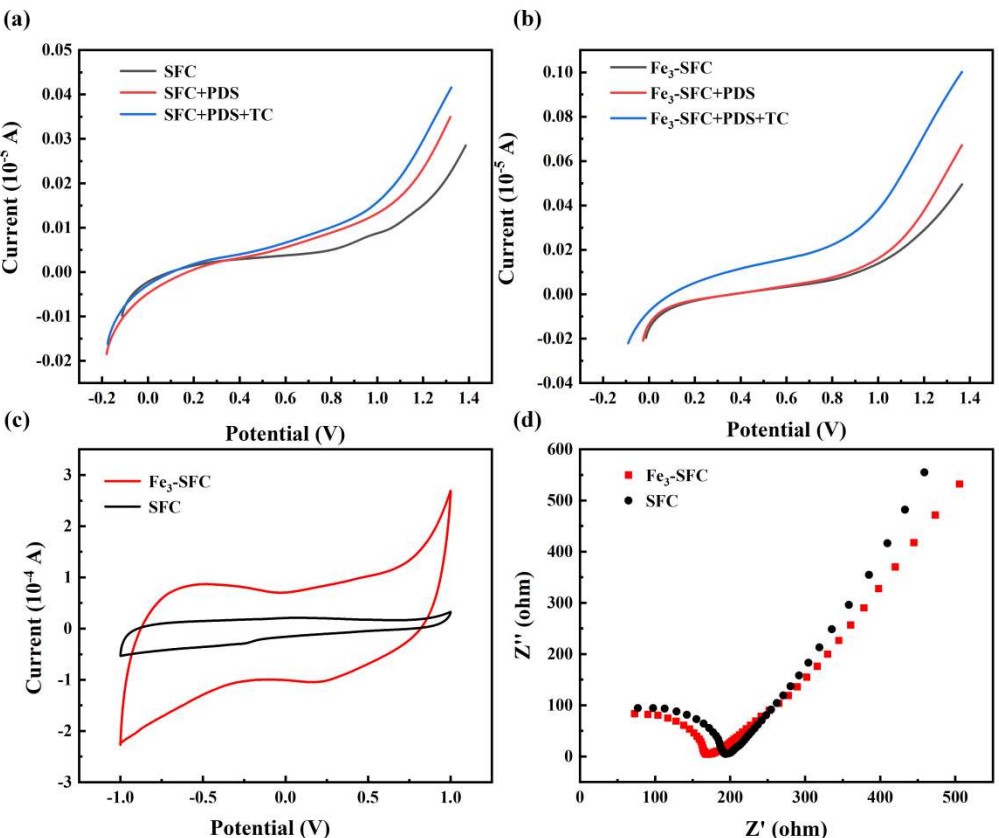

**Figure 10.** LSV curve of SFC (**a**) and Fe$_3$−SFC (**b**) in different situations; CV curves of SFC and Fe$_3$−SFC (**c**); EIS analysis of SFC and Fe$_3$−SFC (**d**).

From Figure 10c, it is worth noting that closed area of Fe$_3$-SFC in the current potential curve is much larger than that of SFC [86], indicating that Fe$_3$-SFC has a large charge storage capacity to accept electrons [87]. The results of LSV and CV further confirmed that catalytic performance of Fe$_3$-SFC for PDS activation was higher than SFC.

EIS tests were performed to further investigate electron transfer ability of two carbon-based catalysts (Figure 10d). The electron transfer ability of biochar-based catalysts is described according to the diameter of the semicircle [88]. The semicircle diameter of Fe$_3$-SFC is significantly smaller than that of SFC. The impedance order of biochar is Fe$_3$-SFC < SFC. This is very consistent with their catalytic performance results. It can be concluded that the incorporation of Fe may improve the electron density and promote the electron flow. In the original structure, doping Fe atoms will help with electron triggering and transfer [89].

In order to facilitate electron transfer between the catalyst and the PDS molecule and facilitate the degradation of organic pollutants, catalytic activation aims to weaken and break down the superoxide O-O link in PDS [61]. XPS spectra of Fe$_3$-SFC before and after the reaction reveal the activation mechanism of PDS (Figure 11). Figure 11a shows that the content of C-C/C=C functional groups in Fe$_3$-SFC decreased from 65.3% to 50.4% after the reaction, which indicated that C-C/C=C was the main reactive functional group to activate PDS [74]. The content of pyridinic N decreased by 18.75% after the reaction, as shown in Figure 11b, indicating that pyridinic N was involved in the activation of PDS [58]. Although graphite N did not change much, previous studies had proved that graphite N could accelerate the transfer of electrons between adjacent ones and destroy the inertia of networks of C, thereby increasing the positive charge of C [61], which was beneficial to pass. PDS was activated by a positively charged nucleophilic reaction that produces

$^1O_2$, implying that graphite N was involved in the process. After the reaction, the O-Fe in Fe$_3$-SFC vanished (Figure 11c), which indicated that O-Fe was the main reactive site when Fe$_3$-SFC catalyzed PDS degradation of TC [74]. After the reaction, the Fe$^{2+}$ bond in the XPS of Fe$_3$-SFC (Figure 11d) disappeared and the Fe$^{2+}$ $2p_{1/2}$ bond area decreased, demonstrating that Fe$^{2+}$ served as a PDS-activating functional group and was involved in the degradation of TC, which further confirmed that Fe$_3$-SFC was involved in the reaction of ferrous iron [76]. The reason why Fe$^{2+}$ $2p_{1/2}$ still exists at this time is that high-energy Fe$^{2+}$ is more likely to react [90].

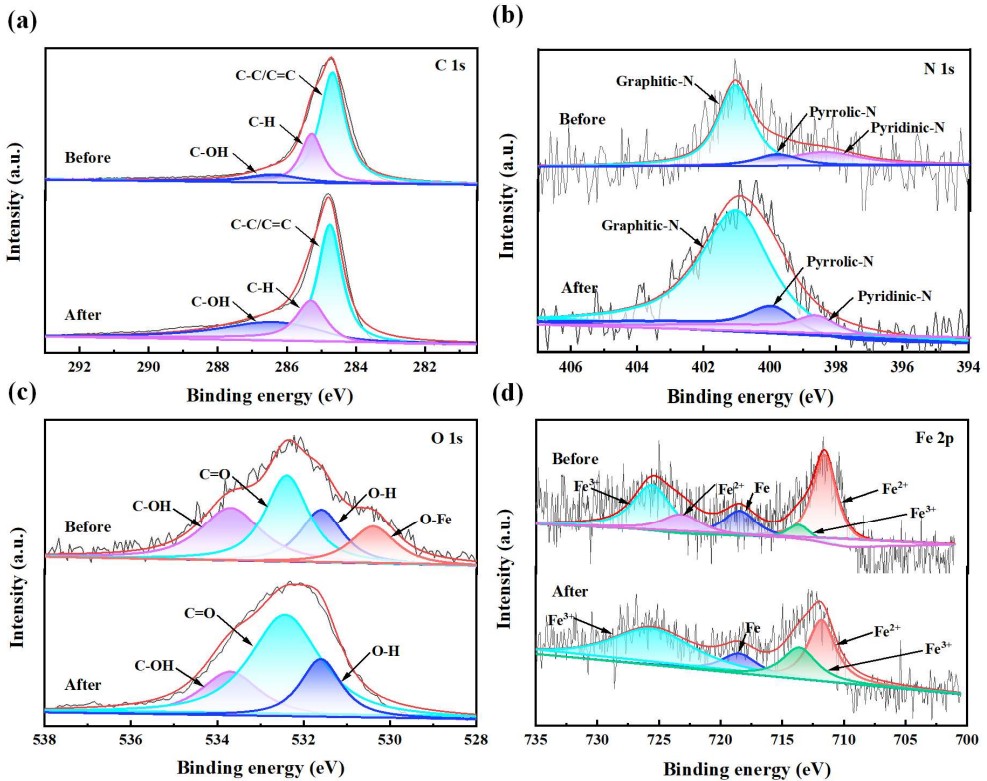

**Figure 11.** XPS spectra of C1s (**a**); O1s (**b**); N1s (**c**) and Fe2p (**d**) of Fe$_3$-SFC before and after reaction.

Based on the above analysis, the radical (SO$_4^{\bullet-}$ and $\bullet$OH) and non-radical ($^1O_2$ and electron transfer) pathways were used to degrade TC in the Fe$_3$-SFC/PDS system, with the latter playing a dominant role (Figure 9a). O-Fe in Fe$_3$-SFC disappeared after the reaction, according to the O1s-XPS spectra of Fe$_3$-SFC before and after the reaction, showing that O-Fe was the primary reaction site [91]. Relevant studies had shown that the content of active ingredients in magnetic BC, especially Fe (II), was closely related to its activation properties. The Fe$^{2+}$ bond in the XPS of Fe$_3$-SFC vanished after the reaction, demonstrating that Fe$^{2+}$ may have acted as a PDS-activating functional group to encourage $^1O_2$ production and was implicated in the degradation of TC. Furthermore, $^1O_2$ was produced by the nucleophilic reaction between the three positively charged C ions surrounding graphite N and PDS. In addition to active species, TC degradation may result from the electron transfer (Figure 10b). Rapid TC degradation might be caused by $^1O_2$ and electron transfer in the Fe$_3$-SFC/PDS oxidation system, whereas $\bullet$OH and SO$_4^{\bullet-}$ played an adjuvant function in TC degradation.

The reversible transformation of Fe$^{2+}$ and Fe$^{3+}$ is shown in Figure 12, and the main reactions involved are [92]:

$$Fe^{3+} + S_2O_8^{2-} \rightarrow Fe^{2+} + S_2O_8^{-\bullet} \tag{11}$$

$$Fe^{3+} + S_2O_8^{2-} \rightarrow SO_4^{-\bullet} + Fe^{2+} + SO_4^{2-} \tag{12}$$

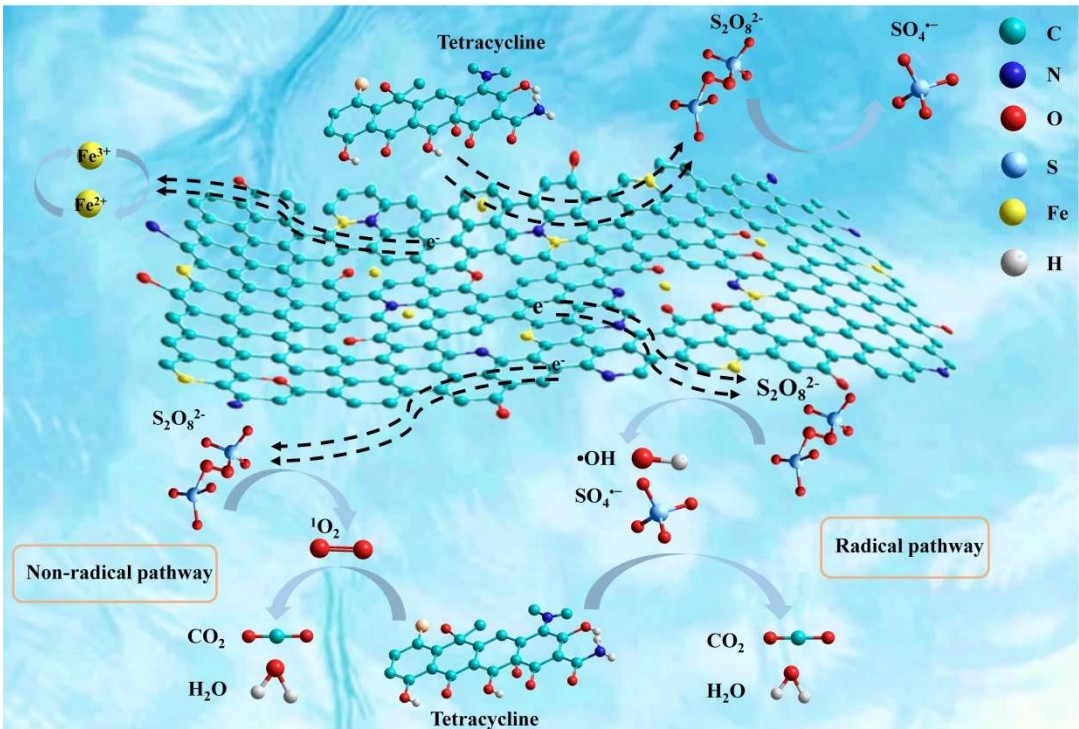

**Figure 12.** Mechanism of the degradation of TC in $Fe_3-SFC/PDS$ system.

In the experiments of Xie et al. [93], the Fe concentration in wastewater discharge after Fe-doped catalyst was degraded by TC was lower than the standard limits from environmental requirements for China's surface water (GB 3838-2002). In this paper, Fe-modified catalysts containing less Fe are used, and the Fe content of the degraded water is low. This will not pollute the environment in a secondary way.

## 4. Conclusions

In this research, using ferric chloride and sterculia foetida as raw materials, a novel magnetic biochar ($Fe_3$-SFC) was successfully synthesized as a persulfate activator to efficiently degrade TC. The characterization results showed that the $Fe_3$-SFC surface was significantly coated with iron corrosion products but less content, and the surface's low-valent Fe (II) on the surface participated in the activation of PDS and was converted into high-valent Fe (III). When the dose of $Fe_3$-SFC is increased, the efficiency of TC's degradation increases, and high TC concentrations will have low degradation efficiencies. PDS concentration (0.5 to 1.5 g/L), solution pH (3.01 to 11.03) and reaction temperature (25 to 45 °C) have little influence on TC degradation. In the presence of $HPO_4^{2-}$ and $HCO_3^-$, the TC degradation in the $Fe_3$-SFC/PDS system was restrained. The degradation mechanism of TC in the $Fe_3$-SFC/PDS system revealed that non-radical pathways involving $^1O_2$, electron transfer, and Fe (II) played a significant role. At the same time, there are some phenomena such as low preparation yield, poor magnetic properties and low recovery rate of $Fe_3$-SFC observed in this experiment. Therefore, the $Fe_3$-SFC synthesized in this study has important practical significance in PS-activated degradation of TC.

**Author Contributions:** Conceptualization, Y.Z., X.J., Z.K. and Z.H.; writing—original draft preparation, Y.Z.; writing—review and editing, Y.Z., X.J., Z.K. and X.K.; visualization, X.K., M.G. and Z.H.; supervision, C.W., J.W. and Z.H.; project administration, C.W., D.Z. and Z.H.; funding acquisition, Z.H. All authors have read and agreed to the published version of the manuscript.

**Funding:** This work was supported by the Hebei National Science Fund for Distinguished Young Scholars (No. E2019209433), Youth Talent Program of Department of Education of Hebei Province (No. BJ2018020) and Hebei Province High-level Talents Funded Project (No. B2020003030).

**Institutional Review Board Statement:** Not applicable.

**Informed Consent Statement:** Not applicable.

**Data Availability Statement:** No data reported.

**Conflicts of Interest:** The authors declare no conflict of interest.

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
