# Peer review of "Degradation of Tetracycline in Water by Fe-Modified Sterculia Foetida Biochar Activated Peroxodisulfate"

_sustainability, doi:10.3390/su141912097_

Round 1

Reviewer 1 Report

The Authors very well discussed the degradation of tetracycline in water by sung Fe modified biochar. the overall manuscript is very good and detailed explanation was done by using different analytical techniques and compared the degradation results with reported literature. this paper is suitable for sustainability journal. 

Reviewer 2 Report

Comments:

0. Major revision. 1. The novelty of this research should be inserted in the text clearly. 2. The advantages and disadvantages of the synthesized Fe-modified sterculia foetida biochar as a catalyst should be investigated. 3. The stability of the synthesized catalyst after the degradation process should be presented by XRD. 4. The ion leaching from the synthesized catalyst during the degradation process should be studied. 5. The “introduction” and “results and discussion” sections of the manuscript can be strengthened and supported with some papers related to the literature: Applied Catalysis B: Environmental 268 (2020), 118443; Journal of Chemical & Engineering Data 55 (2010), 4638-4649; Industrial crops and products 42 (2013), 119-125; Desalination and Water Treatment 47 (2012), 322-333; Fibers and Polymers 11 (2010), 234-240; Journal of Industrial and Engineering Chemistry 20 (2014), 2745-2753; Biotechnology and Bioprocess Engineering 20 (2015), 109-116; Water, Air, & Soil Pollution 224 (2013), 1612; Composites Part B: Engineering 154 (2018), 388-409; Environmental monitoring and assessment 186 (2014), 5595-5604; Inorganica Chimica Acta 487 (2019), 169-176; Journal of Applied Polymer Science 122 (2011), 1489-1499; Journal of Environmental Chemical Engineering 7 (2019), 103243.    

Reviewer 3 Report

This manuscript introduces a novel magnetic biochar (Fe3-SFC) synthesized as a persulfate activator to efficiently degrade TC. Generally, this manuscript lacks innovation and exists some problems.

1.     In the SEM images, how introducing a small amount of Fe changes the micromorphology of SFC? Does this change in morphology also have an effect on the catalytic activity of the material?

2.     In the FR-IR analysis, what kind of interaction occurs between the functional group whose content changes before and after the reaction and TC?

3.     In the EDS analysis, the EDS mapping of Fe element has low contrast, and the size of Figure 2i is too small to be clearly observe.

4.     In the XRD pattern, the peak intensity of the (002) crystal plane decreases and the position of the peak shifts to a high angle. What is the reason for this phenomenon?

5.     In the XPS spectra before and after the reaction, why does Fe2+(2p3/2) still exist, but Fe2+(2p1/2) disappears?

6.     In Figure 12, the conversion between Fe2+ and Fe3+ is a reversible process. What specific reactions will take place during the reaction?

Round 2

Reviewer 2 Report

Accept